# Carbon Nanotube Films with Fewer Impurities and Higher Conductivity from Aqueously Mono-Dispersed Solution via Two-Step Filtration for Electric Heating

**DOI:** 10.3390/nano14110911

**Published:** 2024-05-22

**Authors:** Yingying Chu, Ling Sun, Jing Wang, Zhaoyang Han, Chenyu Wei, Changbao Han, Hui Yan

**Affiliations:** 1Key Laboratory of Advanced Functional Materials, Institute of Advanced Energy Materials and Devices, Ministry of Education, Faculty of Materials and Manufacturing, Beijing University of Technology, Beijing 100124, China; chuyingying@emails.bjut.edu.cn (Y.C.); wangj@emails.bjut.edu.cn (J.W.); hanwayne@emails.bjut.edu.cn (Z.H.); weichenyu@emails.bjut.edu.cn (C.W.); cbhan@bjut.edu.cn (C.H.); hyan@bjut.edu.cn (H.Y.); 2Beijing Guyue New Materials Research Institute, Beijing University of Technology, Beijing 100124, China

**Keywords:** carbon nanotube dispersion, vacuum filtration, filter membrane pore size, electrothermal properties

## Abstract

With the intensification of global climate problems, electric heating has recently attracted much attention as a clean and low-carbon heating method. Carbon nanotubes (CNTs) are an ideal medium for electric heating applications due to their excellent mechanical, electrical, and thermal properties. The preparation of electrothermal films based on an aqueous CNT dispersion as a raw material is environmentally friendly. However, in the traditional one-step filtration method, the residual excess dispersant and the small aspect ratio of the CNTs in the preparation process limit the performance of electrothermal CNT films. In this paper, we report a two-step filtration method that removes the free dispersant and small CNTs in the first filtration step and obtains denser CNT films by controlling the pores of the filter membrane in the second filtration step. The results suggest that, compared to the CNT_1_ film obtained from one-step filtration, the CNT_1-0.22_ film, obtained from two-step filtration using 1 and 0.22 μm membranes, has a smoother and flatter surface, and the surface resistance is 80.0 Ω sq^−1^, which is 29.4% lower. The convective radiation conversion efficiency of the CNT_1-0.22_ film is 3.36 mW/°C, which is 36.1% lower. We anticipate that such CNT films could be widely applied in building thermal insulation and underfloor heating.

## 1. Introduction

In recent years, electric heating has gradually permeated daily life, in line with the continuous progress in science and technology and improvements in environmental awareness [1,2]. Compared with metal-based electric heating materials, carbon-based materials have received widespread attention due to their advantages of high-temperature resistance and stability and high electric conversion efficiency [3,4]. Such electric heating materials include graphite [5], carbon black [6], graphene [7], and carbon nanotubes [8]. Graphite and carbon black have high resistance, while CNTs are considered ideal electrothermal materials for their light weight, good toughness, and excellent electrothermal properties [9,10,11,12,13]. However, due to their high surface energy, CNTs are prone to agglomeration, and it is difficult to directly exploit their nano advantages in applications. Therefore, when industrially preparing CNT dispersions, adding a certain proportion of dispersant is necessary to improve CNT dispersibility in solvents [14,15,16]. Preparing highly conductive CNT films using low-carbon and environmentally friendly aqueous CNT dispersions is an important development direction in this field [17,18,19].

The prerequisite for the preparation of such films is to obtain highly dispersed CNTs for highly conductive network construction while minimizing the residues of the dispersant used. In practice, the dispersant is often disproportionately used to guarantee a relatively stable dispersion. As a result, firstly, a large amount of free dispersant inevitably occurs in the CNT dispersion, and it influences the final electrical conductivity of the films [20]; secondly, the external treatments used in the dispersion process, such as high-speed stirring and grinding, significantly reduce the intrinsic size of the CNTs, lowering the aspect ratio and increasing the number of junctions. Thus, the networking of the CNTs hinders the transfer of electrons in the films [21,22,23]. Notably, these small, highly dispersed CNTs can form poorly conductive electric paths when encapsulated in the films produced, and they need to be sieved out. These impurity problems strongly limit the utilization of electrothermal films based on CNT dispersions.

Vacuum filtration is a method commonly used to prepare electrothermal CNT films [24,25]. It can remove the free dispersant molecules/ions and the CNTs with small aspect ratios from the system through filtration by confining the membrane’s pore sizes [26,27]. However, the traditional one-step filtration cannot solve the impurity problems (free dispersant and small CNTs) simultaneously and spontaneously. Direct one-step filtration using a filter membrane with a large pore size removes both the free dispersant and small CNTs, but this type of CNT membrane has a low density and high surface resistance. Direct one-step filtration using a filter membrane with a small pore size does not remove smaller CNTs either, and this CNT membrane is brittle. Herein, we present a two-step filtration method for the preparation of CNT films. A membrane with a larger pore size (filter membrane with pore size α_1_ < the length of CNTs) is first used to remove the free dispersant and the CNTs with a relatively small aspect ratio, and then a filter membrane with a smaller pore size (filter membrane with pore size α_2_ << α_1_) is used to obtain denser CNT films from the long CNTs after the looser films are redispersed. The resulting CNT film has good electrothermal properties.

## 2. Materials and Methods

### 2.1. Materials

CNT dispersions were purchased from Beijing Carpo Co., Ltd. (Beijing, China). Nylon filter membranes were purchased from the Haining Yibo filter material factory (the tolerance temperature was 120 °C). Polyethylene terephthalate (PET) substrates were purchased from Flying Optoelectronics (the critical point temperature was 150 °C).

### 2.2. Preparation of CNT Films

The CNT films were prepared via the vacuum filtration method. The vacuum pump’s pressure value was −0.1 MPa, and a polytetrafluoroethylene agitator blade was used for stirring the CNT dispersions at 300 rpm. The two-step fabrication process of the CNT films is illustrated in Figure 1. Firstly, 100 mL of the CNT dispersions was filtrated through a 1 µm nylon filter membrane. Subsequently, the CNTs were re-dispersed in 100 mL deionized water through ultrasound for 20 s, and the CNT solution was again filtrated through a 0.22 µm nylon filter membrane. Finally, the prepared CNT_1-0.22_ films were dried at 60 °C for 4 h. In the control group, 100 mL of the CNT dispersion was directly filtrated through a 1 µm nylon filter membrane, and then the prepared CNT_1_ films were dried at 60 °C for 4 h.

The two ends of the 3 mm wide and 20 mm long CNT film were fixed to a polyethylene terephthalate (PET) substrate with conductive adhesive, and then a copper wire was fixed to both ends of the CNT film with conductive silver paste. Next, the conductive silver paste was cured at 60 °C for 1 h in the oven for electrothermal performance testing.

### 2.3. Characterization

The size and the dispersion state of the CNT dispersions and the surface and cross-section images of the CNT films were characterized using a cold-type field emission scanning electron microscope (SU-9000, Hitachi, Tokyo, Japan). The CNT dispersion’s morphology was characterized using an atomic force microscope (MultiMode 8, Bruker, Saarbrücken, Germany). The size of the CNT dispersion was characterized using a zeta potential particle size analyzer (Zetasizer Nano ZS90, Malvern, Malvern City, UK). The 3D structure and roughness of the surfaces of the CNT films were characterized using a confocal laser scanning microscope (Olympus, Tokyo, Japan). The structural features of the pristine CNT dispersions and CNT films were identified using a high-resolution Raman spectrometer (Renishaw, London, UK). The tensile strengths of the CNT film with a gauge length of 20 mm were measured on a Heng Yuan Technology Universal Material Mechanics Testing Machine at a crosshead speed of 1 mm min^−1^. The contact angles of the CNT films were measured using a contact angle meter (Dataphysics-TP50, Filderstadt, Germany). The surface resistance of the CNT films was measured using a four-probe tester (RTS-8, Guangzhou, China). The ultraviolet spectrogram of the filtrate was obtained using an ultraviolet spectrophotometer (TU-1950, Beijing, China). The electrical properties of CNTs films were investigated by obtaining voltage-time, current-time, and current-voltage curves with a sourcemeter (2400, Keithley Instruments Inc., Cleveland, OH, USA). The temperature time curve of CNT films was recorded using an infrared camera (Fotric 326C, Shanghai, China).

## 3. Results and Discussion

### 3.1. Mechanism of Two-Step Filtration

The SEM, TEM, and AFM characteristics of the CNT dispersions are illustrated in Figure 2a–c. The dispersion of the CNTs was good, and the CNTs had a length of 0–2 μm and a diameter of 10 nm. The CNTs were multiwalled, with amorphous carbon present around the walls, presumably due to the dispersant loaded on the CNTs. Filtration with a filter membrane with a large pore size of 1 μm first removed the free excess dispersant and small CNTs (Figure 2d–f). The particle size distribution of the CNT dispersions was shifted to larger sizes (Figure 2d), and the average particle size of the CNT dispersions increased from 163.4 nm to 204.3 nm (Figure 2e) [28]. In the second step of filtration, using a filter membrane with a small pore size of 0.22 μm, the average particle size of the CNT dispersion was almost unchanged, changing from 204.3 nm to 204.9 nm. In Figure 2f, the UV–vis spectra of the filtrates are shown. It can be seen that the CNTs absorbed throughout the whole UV–vis wavelength band, and the absorption intensity was stabilized at one value, without characteristic absorption peaks (the CNT solution was obtained via the ultrasonication of a CNT powder dispersed in water for 30 min). The deionized water passing through the nylon filter membrane had a characteristic absorption peak at 203 nm, presumed to be due to water passing through the nylon filter membrane, carrying the macromolecules in the membrane. Three peaks appeared in the one-step-filtered filtrate: the characteristic absorption peaks of the dispersant at 226 nm and 267 nm and the characteristic absorption peak of the deionized water filtered through the nylon filter membrane (203 nm). The second filtration step was carried out using a filter membrane with a small pore size of 0.22 μm, and the UV–vis spectra show peaks only at 203 nm, without peaks related to the dispersant. The peak intensity was close to 0 near the 800 nm wavelength, indicating that there were no CNTs or dispersant in the filtrate. To determine whether the small-pore-size filter membrane adsorbed the dispersant, in the second filtration step, a membrane with a large pore size of 1 μm was used, and the absorption intensity near 800 nm was relatively small, indicating that the small CNTs were removed in the first filtration step, and the filtrate had fewer CNTs. The characteristic absorption peak appeared at 203 nm, and the shoulder peak was close to 267 nm. This was attributed to the small proportion of CNTs in the filtrate and the dispersant loaded on the carbon nanotubes. Therefore, the first filtration step using a 1 μm membrane at this concentration completely removed the free dispersant and the small CNTs.

### 3.2. Characterization of CNT Films

Figure 3a is a physical diagram of the CNT film. In the one-step filtration using a filter membrane with a large pore size, a fraction of the CNTs were adsorbed into the nylon filter membrane. In Figure 3b, the black CNTs can be seen from the back of the CNT_1_ film. This in-plane and out-of-plane stacking created a heavily undulating surface (Figure 3d,g,h), the contact between CNTs was not close. In the second filtration step using a small-pore-size filter membrane, the CNTs could completely cover the pore size and were almost completely retained (Figure 3c), and the surface was relatively flat (Figure 3e,i,j). The SEM images of the CNT_1_ film (Figure 3d) and CNT_1-0.22_ film (Figure 3e) were analyzed using ImageJ (Figure 3f) [29,30]. The average pore diameters of the CNT_1_ film and CNT_1-0.22_ film were 41.0 nm and 32.4 nm, respectively; the percentages of the pore area were 45.6% and 41.1%, respectively, which corresponded, to a certain extent, to the fact that the CNT_1-0.22_ film was denser. Therefore, denser CNT films (Figure 3d–f) constructed by long CNTs (Figure 2d,e) which were obtained using this two-step filtration method.

As shown in Figure 4, 2D planar and 3D stereograms of the CNT_1_ and CNT_1-0.22_ films were obtained using laser confocal microscopy, again reflecting that the CNT_1-0.22_ film obtained from the two-step filtration method possessed a flatter surface. The surface roughness of the CNT films was measured using a confocal laser scanning microscope [31,32]. In the roughness measurement, Sa represents the arithmetic average height of the measured area, which is essential evidence of the roughness of the sample. Sz denotes the maximum height in the measured area. The parameters measured for the CNT_1_ film were Sa = 0.898 µm and Sz = 19.444 µm, and the parameters measured for the CNT_1-0.22_ film were Sa = 0.572 µm and Sz = 15.162 µm. The CNT_1-0.22_ film had lower surface roughness and a smooth, flat surface. The I_D_/I_G_ values of the CNT dispersions, CNT_1_ film, and CNT_1-0.22_ film were 1.22, 1.20, and 1.17. The I_D_/I_G_ value of the CNT_1-0.22_ film was the smallest, indicating the minor defects and regular stacking of the CNT_1-0.22_ film (Appendix A). A smooth surface is less prone to the accumulation of dust and dirt. At the same time, it is conducive to airflow and can improve the heat dissipation efficiency. The stress–strain curve of the CNT_1-0.22_ film at a tensile strength of 16 MPa is shown in Appendix A.

### 3.3. Hydrophilic Properties of CNT Films

The raw material used in this study was an aqueous CNT dispersion solution. As shown in Figure 5a,e, after the droplets were released onto the surfaces of the CNT films, the contact angles of the CNT_1_ film and CNT_1-0.22_ film were 88.6° and 89.3°, respectively, and the surfaces showed a certain degree of hydrophilicity. The large porosity of the CNT_1_ film allowed the liquid to easily enter the interior of the CNT_1_ film. At the same time, the rough structure of the CNT_1_ film increased the resistance of water droplet diffusion and led to a relatively slow dispersion speed (Figure 5i). As shown in Figure 5a–d, after 40 s, 80 s, and 150 s, the contact angles of the water droplets were 58.2°, 43°, and 0°, respectively. The surface of the CNT_1-0.22_ film was relatively smooth, and the water droplets diffused rapidly on its surface (Figure 5e–h,j). The CNT_1-0.22_ film was not easily wetted when the CNT film was placed in the air for a long time, and if water adhered to the surface of the CNT film, it could be diffused and evaporate quickly.

### 3.4. Electrical Properties of CNT Films

As shown in Figure 6, the surface resistance of CNT_1-0.22_ was 80.0 Ω sq^−1^, which is lower than the value of 113.3 Ω sq^−1^ for the CNT_1_ film obtained using one-step filtration. It is also lower than that of the CNT_1-1_ film obtained through two-step filtration using a 1 μm filter membrane, which had a surface resistance of 123.2 Ω sq^−1^. During filtration, the concentration and volume of CNTs need to be controlled to achieve optimal results. As shown in Appendix A, filtration was performed using 100 mL (1 V), 200 mL (2 V), and 300 mL (3 V) of the CNT dispersion. With the increase in the filtration volume, the decrease in the surface resistance of the CNT_1-0.22_ film after two-step filtration became minor, amounting to 29.3%, 6.5%, and 5.6%, respectively. During the filtration process, the pore size of the new network formed between the CNTs became smaller, and it was impossible to filter out more CNTs. The percentage of retained CNTs increased, and the surface resistance of the CNT_1-0.22_ film proportionally decreased; at the same time, the electrical conductivity increased. As shown in Appendix A, the CNT dispersion was filtered using different pore sizes, and the CNT film obtained via small-pore-size filtration had better electrical conductivity. However, because small-pore-size filtration retained more small CNTs, the CNT film formed was more brittle if the number of CNTs was higher. As shown in Appendix A, when filtered with 300 mL of the CNT dispersion, the CNT_0.65-0.22_ film underwent curling and surface embrittlement. Meanwhile, if only a small-pore-size filter membrane is used for one-step filtration, the dispersant is likely to be adsorbed in the basal nylon filter membrane, and the CNT film will exhibit an unstable state.

### 3.5. Thermal Properties of CNT Films

#### 3.5.1. Thermal Properties at Constant Voltage

Appendix A shows a schematic diagram of the assembled CNT heating film and an infrared image of the heat generation. The time-dependent temperature curves of the CNT film in Figure 7 can be divided into three parts: a temperature ramp, a steady-state temperature, and a temperature decay. For the CNT film, the temperature increased quickly when a constant voltage of 10 V was applied at 10 s; it reached a maximum value within ~50 s and decreased quickly to room temperature when the voltage was removed at 360 s.

In the first and third parts, the temperature growth with time can be expressed as in Equation (1), and the temperature reduction with time can be expressed as in Equation (2) [33,34,35].
(1)$$\frac{T_{t}-T_{i}}{T_{m}-T_{i}}=1-\exp\left(-t/\tau_{g}\right).$$
(2)$$\frac{T_{t}-T_{i}}{T_{m}-T_{i}}=\exp\left(-t/\tau_{d}\right).$$
where $T_{i}$ is the initial sample temperature, $T_{m}$ is the maximum sample temperature, $T_{t}$ is an arbitrary temperature at time *t*, $\tau_{g}$ is the characteristic growth time constant, and $\tau_{d}$ the characteristic reduction time constant. For all CNT films exhibiting electric heating behavior, the $\tau_{g}$ values were calculated by fitting the data for the first ramping part, and the $\tau_{d}$ values were calculated by fitting the data for the third ramping part of the time–temperature curves in Figure 7a. The results are listed in Table 1. The average $\tau_{g}$ and $\tau_{d}$ values at 5~10 s for all CNT films indicate that the films exhibited rapid temperature responsiveness to the applied voltages, reaching stable maximum temperatures within a few seconds.

In the maximum temperature zone, according to the energy conservation law, the heat gain by electric power is considered to be emitted to the surroundings of the composite films through radiation and convection. Therefore, the heat transferred through radiation and convection, $h_{\gamma +C}$, can be evaluated with the following equation:(3)$$h_{\gamma +C}=\frac{I_{C}\cdot V_{0}}{T_{m}-T_{i}}.$$
where $I_{C}$ and $V_{0}$ are the steady-state current value and the initial applied voltage, respectively. As a result, the $h_{\gamma +C}$ values were calculated and are listed in Table 1. The average $h_{\gamma +C}$ values of the CNT films were low. The $h_{\gamma +C}$ value of 3.36 mW/°C for the CNT_1-0.22_ films indicates their excellent electric heating efficiency, requiring relatively low electric power to maintain the maximum temperatures.

As shown in Appendix A, a constant voltage of 10 V was applied, and the change in the current of the CNT film was recorded. It can be seen that the CNT film’s heating was relatively stable at the operating voltage of 10 V. The CNT_1-0.22_ film had the lowest resistance and the highest current among them. Figure 7b shows the time-dependent temperature changes at various applied voltages of 3–20 V for the CNT_1-0.22_ film. The steady-state maximum temperatures (T_max_) were found to increase when increasing the applied voltage. Figure 7c,d show the surface morphology of the CNT_1-0.22_ film at a constant voltage of 10 V and 20 V. At a constant voltage of 10 V, the CNT network on the film surface was relatively stable (Figure 7c); when 20 V was applied, the temperature generated by the CNT film exceeded the tolerance temperature of the nylon filtration filter membrane, and, due to the intensification of molecular thermal movement, the nylon substrate crumpled; at the same time, the network of the CNTs in the upper layer was destroyed (Figure 7d). The contact resistance became larger, so the temperature decreased during the temperature equilibrium stage (Figure 7b).

#### 3.5.2. Thermal Properties with Increasing Voltage

As shown in Appendix A, the voltage was increased by 2.5 V for 350 s at each operating voltage. Similar to the heating performance at a constant voltage, the voltage was between 0 and 17.5 V, and the CNT_1-0.22_ film was relatively stable (Appendix A and Figure 8a). According to the equation Q = $UIt=\frac{U^{2}}{R}$*t* = $I^{2}Rt$, the temperature and voltage are quadratically related (Figure 8b). When the voltage was increased to 20 V, the temperature of the CNT_1-0.22_ film exceeded 120 °C, thus exceeding the tolerance temperature of the nylon filter membrane. The nylon filter membrane began to crumple, the contact resistance R became larger, and the rate of the temperature increase with the voltage became smaller (Figure 8b,c). When the temperature exceeded 200 °C, the CNT_1-0.22_ film broke and no longer generated heat. Both the nylon substrate and PET were charred and the surface CNT network became sparse (Figure 8d). Since the CNT_1_ and CNT_1-1_ films’ surface resistance was relatively large, the current was relatively small at the same operating voltage, and the heating temperature was relatively low; that led to a higher voltage under the films’ fracture.

#### 3.5.3. Stable Performance

Figure 9a shows the temperature variation in the CNT_1-0.22_ film under a 10 V cyclic voltage. The experimental results demonstrate that the CNT_1-0.22_ film has good electrothermal repeatability and stability. The CNT_1-0.22_ film was shaped into a 3 × 3 square and then folded into a heart shape, as shown in the photograph in Figure 9b. It can be seen from the time–temperature curve that, at an operating voltage of 10 V, the CNT_1-0.22_ film functioned well, with good flexibility, and the CNT network on the surface was not damaged.

The CNT films studied in this work exhibited good electrothermal properties. As shown in Table 2, the electrical and thermal properties of the CNT films were superior to those of many other CNT films fabricated using different methods reported in the literature. This work provides a simple approach to the fabrication of highly electrothermal CNT films. The fabrication method can be easily scaled up to allow the production and processing of these materials in large batches.

## 4. Conclusions

We propose a two-step filtration method to directly treat commercial CNT dispersions to obtain CNT films with fewer impurities and higher conductivity from an aqueously mono-dispersed solution, for use as high-performance electrothermal CNT films. The method is simple to operate, environmentally friendly, and low-cost. The resistance of the CNT_1-0.22_ film obtained via this two-step method is 80.0 Ω sq^−1^, which is 29.4% lower than that obtained via the one-step filtration method. Under a 10 V operating voltage, the CNT_1-0.22_ film can reach 70 °C within 40 s, with a 45.8% temperature increase compared to the CNT_1_ film. The convective radiation conversion efficiency of the CNT_1-0.22_ film is 3.36 mW/°C, which is reduced by 36.1% compared to the CNT_1_ film. Additionally, the surface of the CNT_1-0.22_ film is smoother, and the roughness is reduced by 57% compared to that obtained via one-step filtration. This facilitates strong heat dissipation, and thus, moisture diffuses and evaporates rapidly from the film’s surface in humid conditions. We believe that such filtration with this two-step modification would be suitable for a battery of nanomaterials, and the current CNT applications can be greatly improved, especially targeting unprecedented environmental crises.

## Figures and Tables

**Figure 1 nanomaterials-14-00911-f001:**
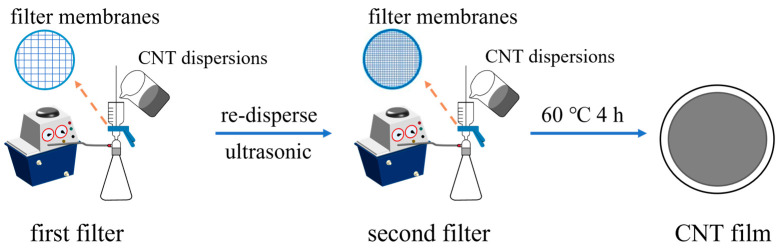
Schematic diagram of the two-step fabrication process of CNT films.

**Figure 2 nanomaterials-14-00911-f002:**
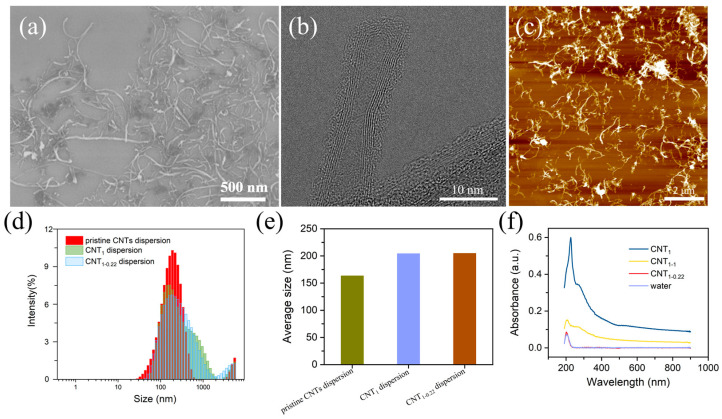
(**a**–**c**) SEM, TEM, and AFM characteristics of CNT dispersions; intensity (%) vs. diameter (nm) (**d**) and average particle size (**e**) of pristine CNT dispersions, CNT_1_ dispersions, and CNT_1-0.22_ dispersions; (**f**) UV–vis spectra of filtrates of CNT_1_ film, CNT_1-1_ film, CNT_1-0.22_ film, water, and CNTs.

**Figure 3 nanomaterials-14-00911-f003:**
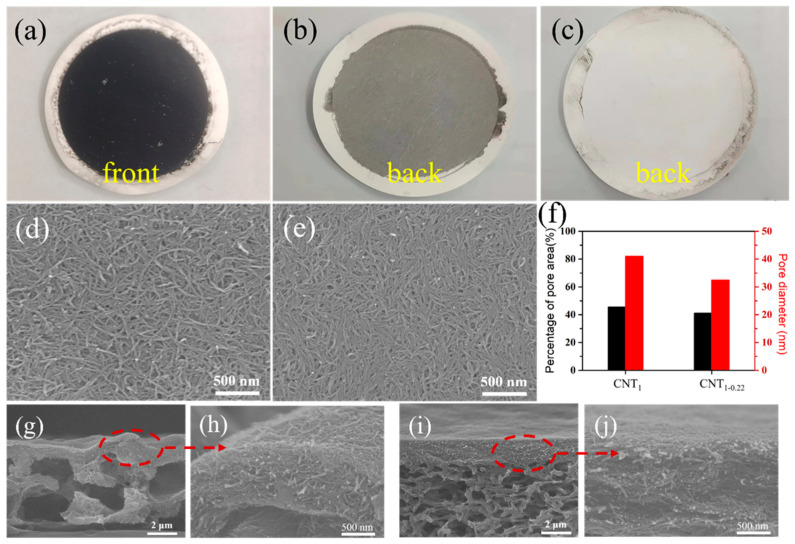
(**a**) Physical diagram of CNT film; back photographs of CNT_1_ film (**b**) and CNT_1-0.22_ film (**c**); surface images of CNT_1_ film (**d**) and CNT_1-0.22_ film (**e**); (**f**) average film pore diameter and percentage of pore area of CNT film obtained via ImageJ2 software; cross-section images of CNT_1_ film (**g**,**h**) and CNT_1-0.22_ film (**i**,**j**).

**Figure 4 nanomaterials-14-00911-f004:**
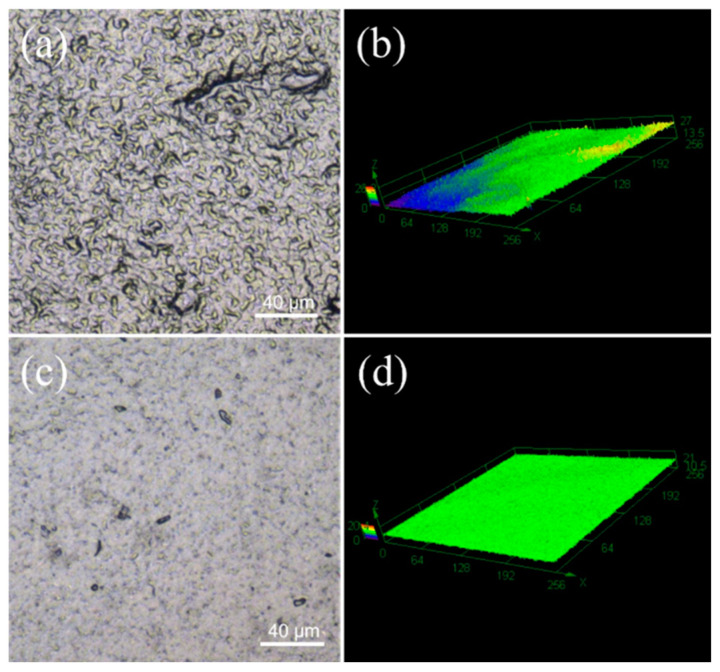
Sample surface microscopic morphology and 3D image of CNT_1_ film (**a**,**b**) and CNT_1-0.22_ film (**c**,**d**).

**Figure 5 nanomaterials-14-00911-f005:**
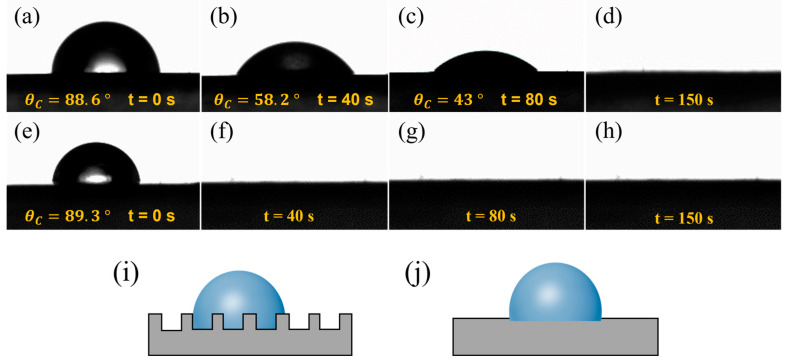
Dynamic contact angles of CNT films at different shooting times. (**a**–**d**) CNT_1_ film and (**e**–**h**) CNT_1-0.22_ film: (**a**,**e**) t = 0 s; (**b**,**f**) t = 40 s; (**c**,**g**) t = 80 s; (**d**,**h**) t = 150 s. (**i**,**j**) Surface wetting state of CNT_1_ film and CNT_1-0.22_ film.

**Figure 6 nanomaterials-14-00911-f006:**
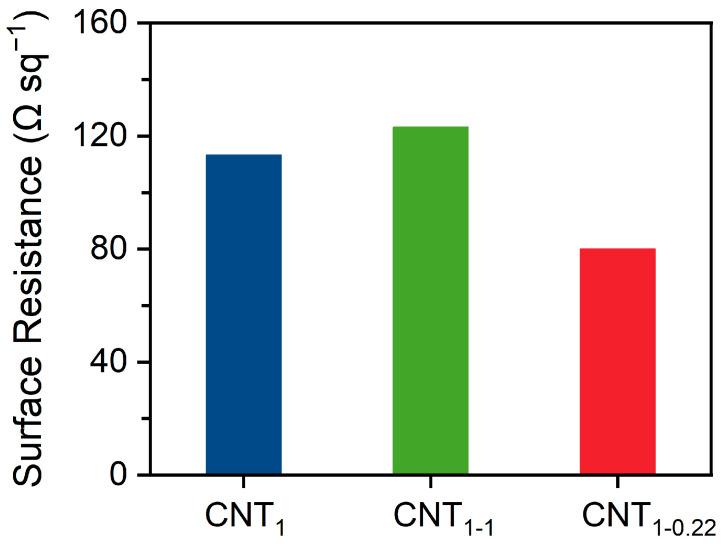
Surface resistance of CNT_1_ film, CNT_1-1_ film, and CNT_1-0.22_ film.

**Figure 7 nanomaterials-14-00911-f007:**
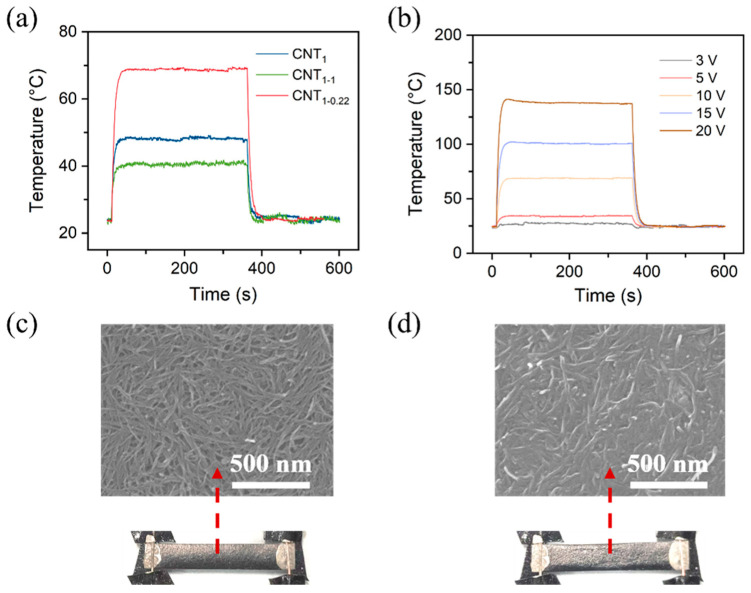
(**a**) Time-dependent temperature changes at applied voltages of 10 V for CNT_1_ film, CNT_1-1_ film, and CNT_1-0.22_ film; (**b**) time-dependent temperature changes at various applied voltages of 3–20 V for CNT_1-0.22_ film. Photographs and SEM images of CNT_1-0.22_ film at a constant voltage of 10 V (**c**) and 20 V (**d**).

**Figure 8 nanomaterials-14-00911-f008:**
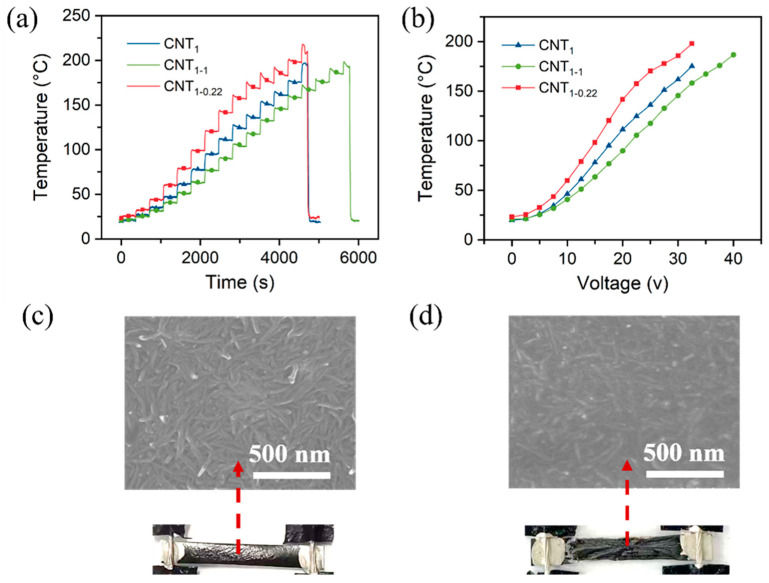
(**a**) Heating performance of CNT films under increasing voltage; (**b**) relationship between temperature and voltage; (**c**) SEM images of CNT_1-0.22_ film at a voltage of 20 V; (**d**) SEM images of CNT_1-0.22_ film at a voltage of 35 V.

**Figure 9 nanomaterials-14-00911-f009:**
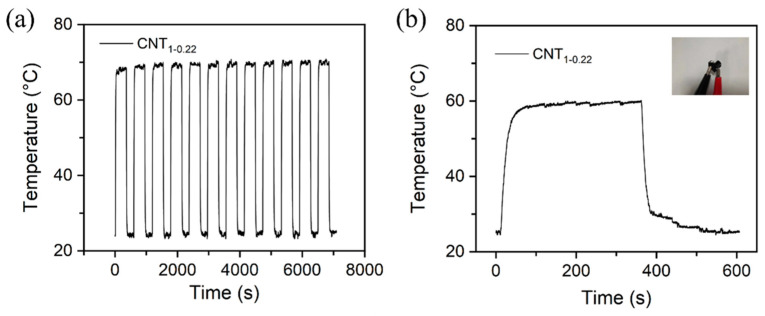
(**a**) Change in temperature of CNT_1-0.22_ film under a cyclic voltage of 10 V; (**b**) change in temperature–time curve for heart-shaped CNT_1-0.22_ film under an applied voltage of 10 V in electric heating experiments.

**Table 1 nanomaterials-14-00911-t001:** Characteristic parameters (τ_g_, τ_d_, and h_r+c_) for the electrical heating behavior of CNT_1_ film, CNT_1-1_ film, and CNT_1-0.22_ film at an applied voltage of 10 V.

	Voltage (V)	$\tau_{g}$ (s)	R^2^	$\tau_{d}$ (s)	R^2^	$h_{\gamma +C}$ (mW/°C)
CNT_1_	10	6.38 ± 0.03	0.99	6.60 ± 0.03	0.93	5.26
CNT_1-1_	10	5.83 ± 0.04	0.97	6.32 ± 0.05	0.88	6.75
CNT_1-0.22_	10	8.21 ± 0.20	0.97	9.4 ± 0.106	0.98	3.36

**Table 2 nanomaterials-14-00911-t002:** The electrothermal properties of the CNT films in this work compared to those of other CNT-based films reported in the literature.

Materials	Thickness (μm)	Resistance (Ω sq^−1^)	Thermal	Ref.
ODA-(MWCNT-COOH/MWCNT-NH_2_)_6_ film	5.6	1160	60 °C at 30 V	[36]
CNT film (CNT suspension 2 wt%)	\	6530	50.3 °C at 35 V	[37]
CNT film (CNT suspension 3 wt%)		20,540	22.8 °C at 35 V	
MWCNT content 0.5 wt%	220	\	35 °C at 10 V	[38]
MWCNT content 0.75 wt%			48 °C at 10 V	
MWCNT content 1 wt%			80 °C at 10 V	
C/CNT-20% fiber	\	\	55 °C at 9 V	[39]
CNT/PMIA paper			230 °C at 20 V	[25]
P-ECS	3.6	340	\	[40]
CNT BP	52	2.6	\	[41]
Graphite film	\	771 ± 69	\	[42]
CNT film		247 ± 8		
Graphite and CNT hybrid films		99 ± 9		
This work	1.2	80	70 °C at 10 V	

## Data Availability

All data are contained within the article.

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
