# Peer review of "Carbon Nanotube Films with Fewer Impurities and Higher Conductivity from Aqueously Mono-Dispersed Solution via Two-Step Filtration for Electric Heating"

_nanomaterials, 2024, doi:10.3390/nano14110911_

Round 1

Reviewer 1 Report

Comments and Suggestions for Authors

In the manuscript “Fewer-impurities CNTs films with higher conductivity from 2 aqueously mono-dispersed solution via two-step filtration for 3 electric heating” the authors proposed an easy and environmental friendly method to obtain fewer-impurities CNTs- based films with high electro thermal properties.

The work reports interesting results for the processing of CNTs, but it could be reconsidered after major revisions, as following suggested:

·         in the Introduction section some results are reported anticipating the data, it could be more suitable removing them from this section and analyse in depth the advantages and limits of one step method, here.

·         Preparation of CNTs films: specify the pressure applied in vacuum filtration; it could be better inserting in this paragraph the description of the deposition of CNTs film on PET substrate, that is reported in the Characterization paragraph, by reporting a detailed procedure.

·         Fig 2 d-e and related comments in the text (lines 118-120): it is not so clear why the particle size distribution of CNTs1increases with respect to the pristine CNTs. The filtration would allow to reduce the size of the filtered material, instead, the graphs in Fig. 2d and 2e seem to show the contrary. reported

·         Lines152,155 report: “the contact resistance was high, and the conductivity was poor”, but is not clear from which measure these aspects result.

·         Line 161 reports: “constructed by long CNTs, which had excellent electrical conductivity, specify to which measures this evidences result.

·         The paragraph 3.3 “The hydrophilic properties of CNTs films” is explained in a confused way, especially, lines 192-201.

·         line 207: correct the measurement unit of the surface resistance, and check throughout the text and graph in Fig. 6

Reviewer 2 Report

Comments and Suggestions for Authors

The manuscript titled Fewer-impurities CNTs films with higher conductivity from aqueously mono-dispersed solution via two-step filtration for electric heating by Chu et al. reports on the synthesis and characterization of CNT-based films for electrical heating applications. The investigated application is attracting attention for clean and low-carbon heating where carbon nanotubes, CNTs, are ideal because of their excellent mechanical, electrical, and thermal properties. The authors have prepared electro-thermal films based on aqueous CNTs dispersion using two-step filtration approach to overcome residual excess dispersant and small aspect-ratio CNTs and achieved denser CNTs films by controlling the pore of the filter membrane. Their experimental findings suggested that compared to CNT film synthesized from the one-step filtration, the CNT film obtained by the two-step filtration is smoother and flatter surface with higher surface resistance. The convective radiation conversion efficiency is also lower for the second approach which is anticipated to be widely applied for building thermal insulation and underfloor heating. Despite an interesting proposition for CNT films conducted in this work, there are several shortcomings including lack of prime novelty, editorial and mechanical deficiencies. There are too numerous to enlist and since it lacks scientific content to the extent that it is justified to be published in this important journal with higher standards. The paper is not acceptable for publication in its current form.

Comments on the Quality of English Language

The extensive English editorial is required.

Round 2

Reviewer 1 Report

Comments and Suggestions for Authors

The authors improved the manuscript accepting all the suggestions. The manuscript could be accepted in present form.